# Distribution and Factors Associated with *Neisseria gonorrhoeae* Cases in Kampala, Uganda, 2016–2020

**DOI:** 10.3390/idr17050132

**Published:** 2025-10-17

**Authors:** Fahad Lwigale, Conrad Tumwine, Reuben Kiggundu, Patrick Elungat, Hope Mackline, Dathan M. Byonanebye, Andrew Kambugu, Francis Kakooza

**Affiliations:** 1Infectious Diseases Institute, Makerere University, Kampala P.O. Box 22418, Uganda; ctumwine@idi.co.ug (C.T.); hmackline@idi.co.ug (H.M.); dbyonanebye@idi.co.ug (D.M.B.); akambugu@idi.co.ug (A.K.); fkakooza@idi.co.ug (F.K.); 2Lifenet International, Kampala P.O. Box 23308, Uganda; patrickelungat007@gmail.com; 3School of Public Health, Makerere University, Kampala P.O. Box 7072, Uganda

**Keywords:** gonorrhoea, *Neisseria gonorrhoeae*, sexually transmitted infection, distribution, factors, epidemiology, Kampala, Uganda

## Abstract

**Background**: Gonorrhoea is a common sexually transmitted infection with serious health consequences if not well-treated. Resistance to common therapeutic agents and limited diagnostics further heighten its burden on sexual and reproductive health. This study determined the positivity level, spatial distribution and factors influencing test positivity for *Neisseria gonorrhoeae* in Kampala, Uganda. **Methods**: Clinical data and urethral swabs were primarily collected from men with urethritis at 10 high-volume surveillance facilities. Laboratory analysis followed conventional microbiology techniques. Statistical analysis was conducted using R 4.4.3. **Results:** Among 1663 participants, 923 (56%, 95% CI: 53–58%) tested positive for *N. gonorrhoeae*, with comparable levels in Kampala divisions. Co-positivity of HIV and *N. gonorrhoeae* ranged from 5–27%. At bivariable analysis, there was a lower risk of testing positive for *N. gonorrhoeae* among participants aged above 24 years. Individuals who never use condoms or infrequently use them were marginally at a higher risk for positivity compared to routine users. Only age was the independent predictor for positivity with *N. gonorrhoeae* (aPR = 0.93, 95% CI: 0.87–0.99, *p*-value = 0.017), with men aged above 24 years being less likely to test positive for *N. gonorrhoeae*. **Conclusions**: Spatial distribution of *N. gonorrhoeae* positivity in Kampala was found not to be significantly influenced by location in any of the five divisions. Public health interventions should be tailored to focus on the high-risk groups such as men aged below 25 years, incorporating targeted education and prevention programs, particularly emphasizing consistent condom use among sexually active individuals to improve sexual and reproductive health in Kampala and greater Uganda.

## 1. Introduction

Gonorrhoea is a sexually transmitted infection (STI) caused by the bacterium *Neisseria gonorrhoeae*. This is primarily transmitted through vaginal, oral and anal sex [1]. The majority of gonorrhoea cases are asymptomatic, especially among women, which limits early detection and management [2,3]. Symptomatic presentation is mostly among men and these present with urethral discharge while women indicate with vaginal discharge [1,4]. Untreated gonorrhoea can cause serious complications such as pelvic inflammatory disease which in turn may result in chronic pelvic pain, infertility, an increased risk of ectopic pregnancy, and neonatal conjunctivitis, and may also increase the risk of acquiring or transmitting human immunodeficiency virus (HIV) [1,2].

In 2020, the World Health Organization estimated that there were 82.4 million *Neisseria gonorrhoeae* cases globally [1]. The estimated global prevalence of gonorrhoea in 2016 was 0.9% and 0.7% among women and men, respectively [5]. The prevalence appears to be higher in Sub-Saharan Africa (SSA), estimated to be 3.28% as reported among women of reproductive age [6]. A systematic review revealed up to 7.9% positivity for *N. gonorrhoeae* in SSA [7]. Much higher positivity levels have been reported among men with urethral discharge syndrome in Kampala, ranging from 51.1–66.4% [8,9]. Additionally, for those attending four STD clinics in and around Kampala, Uganda, the prevalence of *Neisseria gonorrhoeae* was 55% and 15% among men and women, respectively [10].

Significant levels of antimicrobial resistance have been reported among isolates of *Neisseria gonorrhoeae* against commonly used antibiotics [11,12,13]. This is in addition to the limited availability of timely diagnostic tests to guide choice of treatment antibiotics, especially in low- and middle-income countries (LMICs) [3,13]. These factors have jointly exacerbated the impact of gonorrhoea on global sexual and reproductive health.

Efforts have previously been employed to reduce this burden through the Global Health Sector Strategy on sexually transmitted infections, 2016–2021 [14]. Effective STI control is dependent on understanding their distribution in populations [15]. There is limited data about the distribution and influencers of *N. gonorrhoeae* disaggregated by location in Uganda. Understanding this epidemiology is crucial to identify the most affected areas, most at-risk populations and possible risk factors. This facilitates the optimum allocation of scarce resources to design and implement cost-effective disease control and surveillance strategies. This will contribute to the reduction of the burden in communities in line with the global strategies against STIs, contributing to the achievement of the goals for sustainable development [16]. This study determined the positivity levels and factors associated with a positive test for *Neisseria gonorrhoeae* in the five divisions of Kampala, Uganda.

## 2. Materials and Methods

### 2.1. Study Design and Settings

A cross-sectional study design was used. This was a secondary analysis of *N. gonorrhoeae* surveillance data from the Enhanced Gonococcal Antimicrobial Surveillance Program (EGASP) in Uganda. The data collection was conducted between September 2016 to September 2020 in the 10 identified high-volume facilities. They are located within the capital city, Kampala in central Uganda. These included China-Uganda Friendship Hospital (Naguru), Most at Risk Populations Initiative (MARPI) clinic, Murchison Bay Hospital, Luzira Prison Health Centre (HC) IV, Luzira Upper Prison HC III, Infectious Diseases Institute (IDI) Clinic, Kampala Remand Prison HC III, Kiruddu General Hospital, Kawaala HC III and Kisenyi HC IV.

### 2.2. Study Population

This study was based on surveillance among men presenting with urethritis at the 10 healthcare facilities.

### 2.3. Data Collection and Laboratory Procedures

The detailed sample management and analysis methods have been previously described [17]. In brief, standardized procedures were utilized across sites and only men with urethral discharge syndrome attending these sentinel clinics were considered. Various demographic and socio-sexual details were captured from the participants, and urethral swabs were collected. Positive *N. gonorrhoeae* cases were identified using a combination of laboratory test outcomes including Gram stain test and bacterial culture. Detection of Gram-negative intracellular diplococci in primary urethral samples was the minimum requirement to identify a positive case. This was followed up with a microbiology culture test where small translucent colonies on gonococcal-specific media called modified Thayer–Martin agar and chocolate agar incubated between 35–36.5 °C in 5% carbon dioxide were used to characterize positive cases for *N. gonorrhoeae*. Isolated organisms were further identified using a combination of a positive oxidase test and sugar fermentation outcomes alongside known standard organisms for quality control.

### 2.4. Statistical Analysis

Poisson regression with robust variance was done using R 4.4.3 (R Foundation for Statistical Computing, Vienna, Austria) to identify factors associated with *N. gonorrhoeae* positivity. Multicollinearity was assessed and ruled out, then bivariable analysis was conducted for all independent variables at the 20% level. All variables with a *p*-value ≤ 0.2 were considered for multivariable analysis following the backward stepwise elimination approach. Predictors with *p*-value < 0.05 were regarded to have a statistically significant association with *N. gonorrhoeae* positivity.

## 3. Results

We enrolled 1663 participants within the Greater Kampala Metropolitan Area (GKMA) specifically residing in the five (5) divisions that make up Kampala, including Makindye, Rubaga, Central, Kawempe and Nakawa division. The median age of the study participants was 25 (IQR = 22–30). The median number of sexual partners for the study participants was 2 (IQR = 1–2). Of the 109 people with HIV presenting with urethral symptoms, 55 (50%) were found to have gonorrhoea co-infection. The positivity rates were highest at Kisenyi HC IV at 68% (*n* = 282), followed by Naguru hospital at 57% (*n* = 203) (Table 1).

### 3.1. Positivity Levels of N. gonorrhoeae in Kampala

The overall positivity of *N. gonorrhoeae* based on the Gram stain test in the surveyed population (*n* = 1663) was 73%, 95% CI (71–75%) while the bacterial culture test positivity was 56% (923/1663). The Nakawa division had the majority of positive *N. gonorrhoeae* cases at 291 (31.5%) while Central had the least at 75 (8.1%) (Table 2).

Among participants with HIV, the Rubaga division, at 27% (15/55), showed the highest positivity rates while the Central division had the least (Figure 1).

The positivity of *N. gonorrhoeae* among participants with more than one sexual partner was greater in Nakawa division, at 30% (139/469) followed by Rubaga (Figure 2).

### 3.2. Risk Factors for Testing Positive with N. gonorrhoeae

At bivariable analysis, participants with age greater than 24 were at lower risk of positivity for *N. gonorrhoeae*. Participants who never use condoms and those who use them infrequently were more likely to have a positive test for *N. gonorrhoeae* compared to those who always use condoms (Table 3). There was no significant association between a history of exchange of sex for money, number of sex partners, HIV status and *N. gonorrhoeae* positivity. After adjustment for confounders, only age was the statistically significant factor associated with *N. gonorrhoeae* positivity.

## 4. Discussion

Understanding the geographical distribution and influencing factors for STIs is critical for their proper management to lighten the burden imposed on sexual and reproductive health. In this secondary analysis of an urban surveillance cohort of males with urethritis, we found comparable positivity levels across the five divisions of Kampala. The overall positivity of gonorrhoea was as high as 56% among all patients presenting at STI clinics and other facilities in the Kampala surveillance program for *N. gonorrhoeae* and is similar to 66.4% observed in the earlier study [9]. The positivity based on Gram stain test was greater than the observed organism recovery from bacterial culture. This could be due to the fragile nature of the target organism whose isolation from urethral samples could be affected by various factors including prior antibiotic use, delayed processing after sample collection, and fastidious culture requirements leading to lower positivity. The overall observed level of positivity is greater than levels reported earlier within Uganda and elsewhere [10,18,19]. This could be due to the difference in the social-spatial predisposing factors in these areas, with more in Kampala explaining the observed positivity.

The Nakawa division had over 31 cases per 100 participants tested by culture which was slightly higher than other divisions. This could be due to more men having more than one sexual partner than other divisions. Individuals with more than one sexual partner have been previously associated with a higher risk of infection [20]. It could also be because this division houses multiple people outside the city business center including sex workers, and university students who are more likely to engage in high-risk sexual practices that can increase infection transmission. However, the differences in positivity within the five divisions was not statistically significant.

At bivariable analysis, there was reduced risk for *N. gonorrhoeae* positivity among men with age greater than 24 years, and occasional condom use, although the latter was not found to be an independent predictor. However, this association has been reported elsewhere [19]. Condom use has also been a previously known protective factor against STIs [19,21,22]. At multivariable analysis, only age was the independent predictor for *N. gonorrhoeae* infection where those aged greater than 24 years were less likely to test positive. Other studies have also found a younger age (18–25 years) to be more associated with gonorrhoea infection [9,23,24]. This could be due to this being the most sexually active age group increasing their susceptibility to sexually transmitted infections [25,26]. Additionally, it would be expected that individuals who engage in unprotected sex or have multiple sexual partners are at a higher risk of contracting gonorrhoea despite not being a statistically significant association in this scenario. This could be a type II error due to the possibility of limited statistical power to detect this association.

## 5. Limitations

The study was limited by several factors including the target population of only males presenting with urethral discharge syndrome. Despite being ambiguous, female cases could refine the picture of the disease in this urban area. Advanced biochemical and molecular testing could have facilitated higher case detection and confirmation of *N. gonorrhoeae* and related Neisseria species. In addition, the data on HIV status were collected by self-report and over 19% of the men were not aware of, or did not report their HIV status. The clinical outcome data, and the nature of antibiotic treatment taken prior to clinic attendance, were unknown. Similarly, geographic information system (GIS) data were not obtained to enable location mapping of cases observed.

Since sexual orientation was not included in the model, there is a possibility that the spatial effect accounted for by areas with large populations of men who have sex with men (MSM) and associated social-sexual networks were not controlled for. In addition, we did not capture asymptomatic cases which could account for up to 10% in men. Selection bias was noted as data were collected from specific populations of clinic attendees who were not fully representative of the general population, also compounded by other limitations associated with surveillance level data.

## 6. Conclusions

This study observed no significant variations in the positivity of *Neisseria gonorrhoeae* across the five divisions of Kampala, Uganda. The findings show a high overall positivity of 56% among men attending study clinics. Public health strategies can be tailored to focus on high-risk groups such as those aged below 25 years, incorporating targeted education and prevention programs, particularly emphasizing a single sexual partner and consistent use of condoms.

Future studies should investigate the underlying social-spatial factors contributing to the high prevalence of STIs in Kampala. There is a need for research exploring the mechanisms driving the higher susceptibility to infection in specific populations, such as university students, couples, MSMs and other sexual networks. These studies should also consider potential biases in self-reported data, asymptomatic *N. gonorrhoeae*, and aim to develop more accurate assessment methods. Understanding these factors will be crucial for enhancing public health strategies and improving sexual and reproductive health outcomes in the region.

## Figures and Tables

**Figure 1 idr-17-00132-f001:**
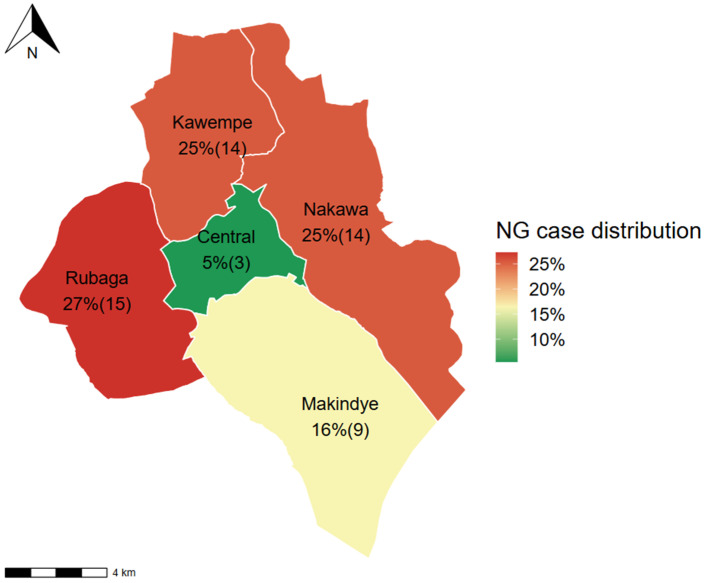
Positivity of *N. gonorrhoeae* among HIV clients in the divisions of Kampala. NG—*N. gonorrhoeae*. Co-positivity of HIV and NG was up to 55 cases.

**Figure 2 idr-17-00132-f002:**
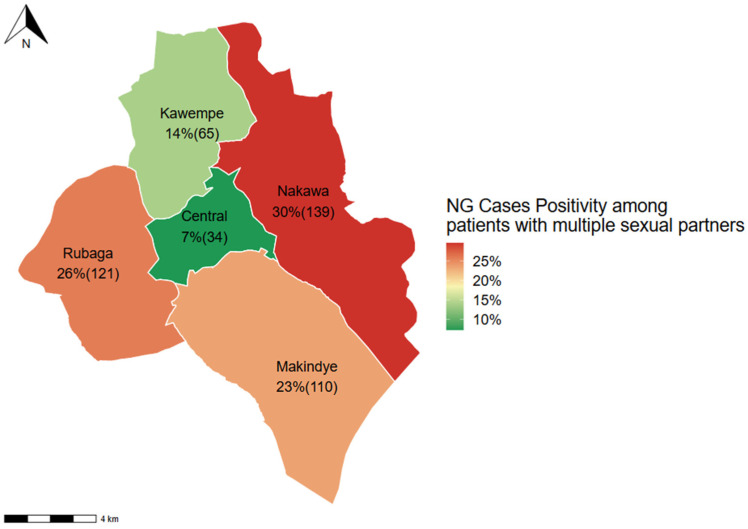
Positivity of *N. gonorrhoeae* among those with more than one sexual partner in Kampala. NG—*N. gonorrhoeae.* Up to 469 cases had more than one sexual partner.

**Table 1 idr-17-00132-t001:** Demographic characteristics of the study participants.

Characteristic	Positive for *N. gonorrhoeae* (Gram Staining) *n* = 1218	Positive for *N. gonorrhoeae* (Gram Staining and Culture) *n* = 923
**HIV Status**		
Negative	874 (71.8%) *	658 (71.2%)
Positive	78 (6.4%)	55 (5.9%)
Unknown	266 (21.8%)	210 (22.7%)
**Age group**		
Median (IQR) age	24 (21, 30) **	
≤24 years	610 (50.0%)	476 (51.6%)
Greater than 24 years	580 (47.6%)	425 (46.0%)
Unknown	28 (2.3%)	22 (2.4%)
**Number of Sex partners in the last six months**		
Median (IQR) no. of sexual partners	2.00 (1.00, 2.00)	
≤1 Partner	553 (45.4%)	412 (44.6%)
>1 Partner	595 (48.8%)	469 (50.8%)
Unknown	70 (5.7%)	42 (4.6%)
**Condom Use**		
Always	10 (0.8%)	8 (0.8%)
Never	463 (38.0%)	336 (36.4%)
Sometimes	722 (59.3%)	565 (61.2%)
Unknown	23 (1.9%)	14 (1.5%)
**Facility Name**		
IDI Clinic	11 (0.9%)	9 (0.9%)
Kampala Remand Prison HC III	43 (3.5%)	26 (2.8%)
Kawaala HC III	189 (15.5%)	147 (15.9%)
Kirruddu GH	150 (12.3%)	115 (12.5%)
Kisenyi HC IV	332 (27.3%)	282 (30.6%)
Luzira Prison HC IV	79 (6.5%)	48 (5.2%)
Luzira Upper Prison HC III	20 (1.6%)	3 (0.3%)
MARPI Mulago	90 (7.4%)	77 (8.3%)
Murchison Bay hospital	37 (3.0%)	13 (1.4%)
Naguru hospital	267 (21.9%)	203 (22.0%)
**History of Sex exchange for Money**	*167 (13.7%)*	*128 (13.9%)*
**Urethral discharge**	*1185 (97.2%)*	916 (99.2%)
**Dysuria**	*1082 (88.8%)*	822 (89.1%)

* Percentage positivity per variable, ** Interquartile range (IQR).

**Table 2 idr-17-00132-t002:** Positivity of *N. gonorrhoeae* within the divisions of Kampala.

Characteristic	Overall n = 1663 (95% CI ^1^)	Central n = 113 (95% CI)	Kawempe n = 218 (95% CI)	Makindye n = 340 (95% CI)	Nakawa n = 579 (95% CI)	Rubaga n = 413 (95% CI)
Positivity (Gram stain)	1218 (73%) (71%, 75%) ^1^	91 (7.5%) (6.1%, 9.1%)	155 (12.7%) (11%, 15%)	244 (20%) (18%, 22%)	409 (33.5%) (31%, 36%)	319 (26.4%) (24%, 29%)
Positivity (Gram stain & Culture)	923 (56%) (53%, 58%)	75 (8.1%) (6.5%, 10%)	113 (12.2%) (10%, 15%)	185 (20.0%) (18%, 623%)	291 (31.5%) (29%, 35%)	259 (28.1%) (25%, 31%)

CI ^1^ = Confidence Interval.

**Table 3 idr-17-00132-t003:** Results from the Bivariable and Multivariable analysis.

Characteristic	Bivariable Analysis (cPR ^1^ (95% CI ^2^))	*p*-Value	Multivariable Analysis (aPR ^3^ (95%CI))	*p*-Value
**Patient Age group**				
≤24 years	1		1	
Greater than 24 years	0.93 (0.88, 0.99) ^2^	0.02 *	0.93 (0.87, 0.99)	0.017 **
**HIV Status**				
Negative	1			
Positive	0.98 (0.87, 0.75)	0.79		
Unknown	1.04 (0.96, 1.14)	0.27		
**Number of sex partners in the last 6 months**				
≤1 Partner	1			
>1 Partner	0.99 (0.93, 1.05)	0.82		
**Condom Use**				
Always	1		1	
Never	1.44 (0.90, 2.29)	0.12 *	1.43 (0.89, 2.30)	0.141
Sometimes	1.48 (0.93, 2.36)	0.09 *	1.47 (0.93, 2.36)	0.115
**History of Sex Money Exchange**				
No	1			
Yes	0.96 (0.88, 1.06)	0.48		
**Division**				
Central	1		1	
Kawempe	0.88 (0.77, 1.01)	0.08 *	0.90 (0.79, 1.03)	0.145
Makindye	0.90 (0.80, 1.02)	0.13 *	0.91 (0.81, 1.03)	0.169
Nakawa	0.89 (1.00, 0.07)	0.07 *	0.90 (0.80, 1.01)	0.07
Rubaga	0.98 (0.88, 1.11)	0.84	0.99 (0.88, 1.11)	0.89

cPR ^1^ = Crude Prevalence Ratios, aPR ^3^ = adjusted Prevalence Ratios; CI ^2^ = Confidence Interval, * = Statistically Significant at *p* < 0.2; ** = Statistically Significant at *p* < 0.05.

## Data Availability

The data presented in this study are available on request from the corresponding author. The raw data supporting the conclusions of this article will be made available by the authors on request.

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
