# Peer review of "Distribution and Factors Associated with Neisseria gonorrhoeae Cases in Kampala, Uganda, 2016–2020"

_2036-7449, 2025, doi:10.3390/idr17050132_

Round 1

Reviewer 1 Report

Comments and Suggestions for Authors

In this manuscript, the authors employed a cross-sectional study design to investigate Neisseria gonorrhoeae (NG) positivity level, spatial distribution of NG infection across geographical divisions in Kampala (Uganda), and factors influencing NG positivity. Overall, the statistical study is clearly described, the design is appropriate, and the conclusions are accurately drawn based on the results. Specific minor comments are provided below.

  • All genus-species names, including those in the title, should be italicized per convention.
  • Provide the country in the background subsection of the Abstract (line 16).
  • The infection gonorrhea does not need to be capitalized within sentences.
  • The word “understating” in line 62 of the Introduction should be revised to “understanding”.
  • Revise line 130 by inserting “one” before “sexual partner”.

Author Response

Response to Reviewer 1 Comments

Thank you very much for taking the time to review this manuscript. Please find the detailed responses below and the corresponding revisions in track changes in the re-submitted files.

Point-by-point response to Comments and Suggestions for Authors

In this manuscript, the authors employed a cross-sectional study design to investigate Neisseria gonorrhoeae (NG) positivity level, spatial distribution of NG infection across geographical divisions in Kampala (Uganda), and factors influencing NG positivity. Overall, the statistical study is clearly described, the design is appropriate, and the conclusions are accurately drawn based on the results. Specific minor comments are provided below.

Comment 1: All genus-species names, including those in the title, should be italicized per convention.

Response: Thank you. We have ensured that genus-species names, including those in the title, are italicized as per convention. This change has been throughout the revised manuscript.

Comment 2: Provide the country in the background subsection of the Abstract (line 16).

Response: The country is added to the updated manuscript abstract.

Comment 3: The infection gonorrhea does not need to be capitalized within sentences.

Comment 4: The word “understating” in line 62 of the Introduction should be revised to “understanding”.

Response: This has been corrected. Thank you for the observation. 

Comment 5: Revise line 130 by inserting “one” before “sexual partner”.

Response: This has been added. Thank you for the observation. 

Reviewer 2 Report

Comments and Suggestions for Authors

Question 1: What exactly was the culture medium used to grow Neisseria gonorrhoeae? This is important to know because it is necessary to ensure that the culture medium has sufficient selectivity to allow only N. gonorrhoeae to grow. Since subsequent identification methods (biochemical tests) are not specific enough, they may also give positive results for other pathogens, e.g. Haemophilus influenzae.

Line 130: The positivity... with more thanone”, the wordoneis missing.

Comments. 

1. Neisseria gonorrhoeae should also be italicized in the title.

2. Determining a patient's positivity based only on Gram staining is not relevant, so samples where no growth occurred on the culture medium should not be taken into account. 

3. Overall, this article does not present relevant data. The authors also point out the limitations of the publication, stating that the individuals examined are not relevant because they were examined in a narrow circle. This is also one of the problems for me, as the 10 healthcare institutions include 3 prison hospitals, which do not represent the actual population. In this case, it would be essential to examine the MSM population and sexual orientation.

I believe that the article could be modified and optimized by using specific indentification tests and specifying the population studied.

Author Response

Response to Reviewer 2 Comments

Thank you very much for taking the time to review this manuscript. Please find the detailed responses below and the corresponding revisions in track changes in the re-submitted files.

Point-by-point response to Comments and Suggestions for Authors

Comment 1: What exactly was the culture medium used to grow Neisseria gonorrhoeae? This is important to know because it is necessary to ensure that the culture medium has sufficient selectivity to allow only N. gonorrhoeae to grow. Since subsequent identification methods (biochemical tests) are not specific enough, they may also give positive results for other pathogens, e.g. Haemophilus influenzae.

Response: Thank you for this comment. The utilised culture media have been added to the recent manuscript version currently reading as “......gonococcal-specific media called modified Thayer-Martin agar and chocolate agar incubated between 35°-36.5°C in 5% carbon dioxide were used….”. Please check line 95-97 in the updated manuscript.

Comment 2: Line 130: The positivity... with more than “one”, the word “one” is missing.

Response: This has been added. Thank you for the observation. 

Comment 3: Neisseria gonorrhoeae should also be italicized in the title.

Response: Neisseria gonorrhoeae has been italicised in the revised manuscript as recommended. Thank you for the comment. 

Comment 4: Determining a patient's positivity based only on Gram staining is not relevant, so samples where no growth occurred on the culture medium should not be taken into account. 

Response: Thank you for this comment.  The overall positivity has been revised to consider positive results from culture testing. The results from Gram staining test approach have been maintained for ease of comparison. Please see Table 1 in the revised manuscript.

Comment 5: Overall, this article does not present relevant data. The authors also point out the limitations of the publication, stating that the individuals examined are not relevant because they were examined in a narrow circle. This is also one of the problems for me, as the 10 healthcare institutions include 3 prison hospitals, which do not represent the actual population. In this case, it would be essential to examine the MSM population and sexual orientation.

I believe that the article could be modified and optimized by using specific indentification tests and specifying the population studied.

Response: Thank you for this comment. The specific identification tests have been added to the methods section as earlier referred to. The study population was clearly mentioned to be men presenting with urethral discharge syndrome in Kampala and managed through the WHO Enhanced Gonococcal Antimicrobial Surveillance Program (EGASP) in Uganda. This is a surveillance program approved with very well-defined criteria and methodology. We believe that the non-inclusion of females or MSM limits generalizability, it is consistent with data suggesting that the major route of STI transmissions in sub-Saharan Africa is heterosexual. Additionally, gonorrhoea incidence is high in prisons compared to the general population and prisoners remain a critical population for gonorrhea/STIs surveillance in most countries[1],[2]. The limitations are to ensure that results are interpreted carefully. This study also focuses on surveillance of drug resistance among people with gonorrhoea and we do not think there is documented differential risk of drug resistance by route of gonorrhea acquisition. Finally, our results provide baseline for future work design especially in case the MSM make part of the study or surveillance objectives.

[1] https://www.cdc.gov/correctional-health/about/index.html

[2] https://ajph.aphapublications.org/action/showCitFormats?doi=10.2105%2FAJPH.2019.305425

Reviewer 3 Report

Comments and Suggestions for Authors

This well written report contains interesting data on the prevalence of gonorrhea, as approximated by looking at EGASP data, in males with urethritis complaints in 5 different districts in Kampala, Uganda. In samples from these men an extremely high prevalence (73%) of Neisseria gonorrhoeae (NG) was found using Gram staining. No factors were found to be significantly associated with the NG prevalence except for age, being higher in men younger than 25 years.

There are major and minor points that should be addressed to further improve this manuscript.

Major comments.

  1. In Table 1 the patient characteristics are shown. In Methods is mentioned that the patient samples were derived from10 high volume test facilities. These facilities vary from STI clinics, hospitals, health centres and a prison. It would be interesting to see how many samples exactly were derived from which test facility, since these facilities very likely represent different populations at risk for sexually infected infections (STI). So I expect that for example a prison population is quite different from a hospital population. Please add these facility data in Table 1.
  2. In Table 1, after ‘Median age’ and after ‘median number of sex partners’ two numbers are given without explaining what these are. Is this the min-max or the range? Please add this in the table.
  3. The above mentioned facility data may also be mentioned in Tables 2 and 3. Please adjust.
  4. The positivity rates of NG are very high using Gram staining. What is the false positivity rate using this technique? The authors state that using bacterial culture a (much) lower rate is found. Please also mention the NG positivity rate using culture in all tables and Figures.
  5. In Table 2 the percentages negative for NG with Gram stain are also mentioned. However, these are fully complementary to the percentages positive and can thus be deleted. Instead it would be better to mention for each district the percentage NG positive as determined using bacterial culture. Please adjust.
  6. In Figures 1 and 2 the colours that are used to show the gradient of NG prevalences is not very well discriminating, going from red via yellow again to red. So now it looks like the red colour for Kawempe is the same as the red colour for Central. Please either use more discriminating colours or do not use any colours at all. The colours do not add anything and just mentioning the numbers is more clear.
  7. In Table 3 is mentioned in the heading that Bivariable and Multivariable analysis is performed. This is not obvious from the legend. Is it so that the ‘cPR = Crude prevalence ratios’ belongs to the bivariate analysis and the ‘aPR = adjusted Prevalence ratio’ belongs to the Multivariate analysis? If so please explain this more clearly in the text.
  8. In lines 137- 139 is mentioned that ‘participants who never use condoms and those who use them infrequently were more likely to have a positive NG test compared to those who always used condoms’. In Table 3 can be seen that this was not significant however. So please add that it was not significantly different (although you may have expected that it would have been significantly different).

Minor comments.

  1. Abstract, line 20: consider using ‘districts’ instead of ‘divisions’ for the different geographical parts in Kampala.
  2. Introduction, line 56: please write Neisseria gonorrhoeae consistently correct throughout the manuscript. The same in line 59 for ‘gonorrhea’.
  3. Line 62: please adjust to ‘populations’
  4. Results, line 130: please add ‘one’ before ‘sexual’.
  5. Discussion, line 145: please replace ‘influencers’ by ‘influences’.
  6. In line 149 ‘at STI clinics’ is mentioned. But actually there were also a lot of other facilities, or is that not so? Please adjust.
  7. Lines 167-168 is mentioned ‘and occasional condom use’. As stated above, this was not significant and should therefor not be mentioned here. You still may say that you would have expected an effect of inconsistent condom use but that this could not be detected in the present study.
  8. Conclusions, line 196: ‘strategies can be tailored to those aged below 25 years’: I actually think that the NG prevalence in all men with urethritis was extremely high, so to discriminate between below and above 25 years is not very helpful.

Author Response

Response to Reviewer 3 Comments

Thank you very much for taking the time to review this manuscript. Please find the detailed responses below and the corresponding revisions in track changes in the re-submitted files.

Point-by-point response to Comments and Suggestions for Authors

Comment: This well written report contains interesting data on the prevalence of gonorrhea, as approximated by looking at EGASP data, in males with urethritis complaints in 5 different districts in Kampala, Uganda. In samples from these men an extremely high prevalence (73%) of Neisseria gonorrhoeae (NG) was found using Gram staining. No factors were found to be significantly associated with the NG prevalence except for age, being higher in men younger than 25 years.

There are major and minor points that should be addressed to further improve this manuscript.

 Major comments.

Comment 1: In Table 1 the patient characteristics are shown. In Methods is mentioned that the patient samples were derived from10 high volume test facilities. These facilities vary from STI clinics, hospitals, health centres and a prison. It would be interesting to see how many samples exactly were derived from which test facility, since these facilities very likely represent different populations at risk for sexually infected infections (STI). So I expect that for example a prison population is quite different from a hospital population. Please add these facility data in Table 1.

Response: Thank you for this comment. The results have further been disaggregated to reveal positivity levels across different surveillance sites. Please see Table 1 in the revised manuscript.

Comment 2: In Table 1, after ‘Median age’ and after ‘median number of sex partners’ two numbers are given without explaining what these are. Is this the min-max or the range? Please add this in the table.

Response: Thank you for this comment. The numbers refer to the Interquartile range(IQR). This has been clarified in the table.

Comment 3: The above mentioned facility data may also be mentioned in Tables 2 and 3. Please adjust.

Response: Thank you for this suggestion. Table 2 and 3 have got values representing the 95% confidence interval and these have been defined as recommended. However, Table 2 is specifically revealing the positivity across the divisions in the study area of Kampala. And Table 3 is revealing outcomes from the regression analysis which is collectively done based on variables collected across the included sites and not individually. This makes it impractical to display the Table 2 and 3 results per facility.

Comment 4: The positivity rates of NG are very high using Gram staining. What is the false positivity rate using this technique? The authors state that using bacterial culture a (much) lower rate is found. Please also mention the NG positivity rate using culture in all tables and Figures.

Response: Thank you for this comment and we agree with your observation.  The overall positivity has been revised to consider positive results from culture testing. The results from Gram staining test approach have been maintained for ease of comparison. Please see Table 1, Figure 1& 2 in the revised manuscript.

Comment 5: In Table 2 the percentages negative for NG with Gram stain are also mentioned. However, these are fully complementary to the percentages positive and can thus be deleted. Instead it would be better to mention for each district the percentage NG positive as determined using bacterial culture. Please adjust.

Response: Thank you for this comment.  The percentage negatives have been eliminated.  NG positive cases using both Gram staining and bacterial culture are added. Please see Table 2 in the revised manuscript.

Comment 6: In Figures 1 and 2 the colours that are used to show the gradient of NG prevalences is not very well discriminating, going from red via yellow again to red. So now it looks like the red colour for Kawempe is the same as the red colour for Central. Please either use more discriminating colours or do not use any colours at all. The colours do not add anything and just mentioning the numbers is more clear.

Response: Thank you for this comment.  The colours are of a heatmap that corresponds to the gradual value change as revealed in the key and some colors could be similar due to closeness of their respective values. However, these have been modified to consider relatively more variable colours. Please check out the figures in the updated manuscript.  

Comment 7: In Table 3 is mentioned in the heading that Bivariable and Multivariable analysis is performed. This is not obvious from the legend. Is it so that the ‘cPR = Crude prevalence ratios’ belongs to the bivariate analysis and the ‘aPR = adjusted Prevalence ratio’ belongs to the Multivariate analysis? If so please explain this more clearly in the text.

Response: Thank you for this comment. The prevalence ratios have been clarified relative to the respective analytical approaches. Please see Table 3 in the revised manuscript.

Comment 8: In lines 137- 139 is mentioned that ‘participants who never use condoms and those who use them infrequently were more likely to have a positive NG test compared to those who always used condoms’. In Table 3 can be seen that this was not significant however. So please add that it was not significantly different (although you may have expected that it would have been significantly different).

Response: Thank you for this comment.  At the bivariable analysis level, the cut-off p-value was ≤ 0.2 and hence this was considered significant. This is why the same variable was considered for further analysis at the multivariable level. However, in the final reporting this is not reported to be a significant association as later confirmed for this scenario.

Minor comments.

 Comment 9: Abstract, line 20: consider using ‘districts’ instead of ‘divisions’ for the different geographical parts in Kampala.

Response: Thank you for this comment.  However, based on the local geographical nomenclature, these subjects are referred to as divisions and not districts. So this had to be maintained.

Comment 10: Introduction, line 56: please write Neisseria gonorrhoeae consistently correct throughout the manuscript. The same in line 59 for ‘gonorrhea’.

Response: Writing the terms as “Neisseria gonorrhoeae” and “gonorrhoea” has been observed throughout the updated manuscript. Thank you for the observation. 

Comment 11: Line 62: please adjust to ‘populations’

Response: This has been corrected in the new manuscript. Thank you for the observation. 

Comment 12: Results, line 130: please add ‘one’ before ‘sexual’.

Response: This has been corrected in the new manuscript. Thank you for the observation. 

Comment 13: Discussion, line 145: please replace ‘influencers’ by ‘influences’.

Response: Thank you for the comment. This was aimed to refer to influencing factors and has been clarified in the current manuscript. Please see line 145 in the new manuscript.

Comment 14: In line 149 ‘at STI clinics’ is mentioned. But actually there were also a lot of other facilities, or is that not so? Please adjust.

Response: Thank you for the comment. This has been generalised to consider all included sites.

Comment 15: Lines 167-168 is mentioned ‘and occasional condom use’. As stated above, this was not significant and should therefor not be mentioned here. You still may say that you would have expected an effect of inconsistent condom use but that this could not be detected in the present study.

Response: Thank you for this comment.  At the bivariable analysis level, the cut-off p-value was ≤ 0.2 and hence this was significant as limitedly mentioned. However, in the final reporting this is clarified not to be a significant association as later confirmed for this scenario at the multivariable analysis level.

Comment 16: Conclusions, line 196: ‘strategies can be tailored to those aged below 25 years’: I actually think that the NG prevalence in all men with urethritis was extremely high, so to discriminate between below and above 25 years is not very helpful.

Response: Thank you for this comment.  The control is indeed applicable to the majority. However, the statement is aimed to favour that group being the most-at risk, based on the observed evidence from the study in case of prioritization especially in resource-limited settings.

Reviewer 4 Report

Comments and Suggestions for Authors

The primary limitation of this study lies in the diagnosis of Neisseria gonorrhoeae based solely on Gram staining and culture. While it is understandable that the research team may have lacked the human or financial resources to perform nucleic acid amplification tests (NAATs), such as urinary PCR, on all participants, this methodological constraint likely led to a substantial underestimation of actual cases.

Furthermore, Gram staining lacks specificity, as it can also detect Neisseria meningitidis, another organism capable of causing urethritis. This may account for the discrepancy observed between Gram-positive findings and culture-positive results reported by the authors.

Another important limitation—acknowledged by the authors—is the absence of data regarding patients’ sexual orientation. As a result, men who have sex with men (MSM) were not screened at all anatomical sites, which is a notable shortcoming. In this population, relying on urine samples alone is insufficient for comprehensive screening.

Additionally, I found the results of the multivariate analyses somewhat surprising. The lack of association between unprotected sex or multiple sexual partners and increased risk of gonorrhea is counterintuitive. The authors should provide further explanation or discuss potential confounding factors that could account for this unexpected finding.

Minor comments:

  • The statement that gonorrhea is “asymptomatic, especially in women” is poorly phrased. While this may be true for women, gonorrhea is predominantly symptomatic in men and the wording should reflect this contrast.

  • Line spacing in the introduction is inconsistent and should be standardized for clarity and presentation.

  • Table 1 is difficult to interpret, particularly the percentages, which are unclear. I recommend reformatting the table to improve its readability and visual coherence.

Author Response

Response to Reviewer 4 Comments

Thank you very much for taking the time to review this manuscript. Please find the detailed responses below and the corresponding revisions in track changes in the re-submitted files.

Point-by-point response to Comments and Suggestions for Authors

Comment 1: The primary limitation of this study lies in the diagnosis of Neisseria gonorrhoeae based solely on Gram staining and culture. While it is understandable that the research team may have lacked the human or financial resources to perform nucleic acid amplification tests (NAATs), such as urinary PCR, on all participants, this methodological constraint likely led to a substantial underestimation of actual cases.

Response: Thank you for this comment. The results were generated through the WHO Enhanced Gonococcal Antimicrobial Surveillance Program (EGASP) in Uganda. This is a surveillance program approved with well-defined criteria and methodology including diagnostic methods. Despite the limitation of relying on culture-based method, it is still a dominant method used in public health programs in resource limited settings such as the study area. Their generated results are considered informative for programs in countries that still rely on these methods due to resource constraints.  

Comment 2: Furthermore, Gram staining lacks specificity, as it can also detect Neisseria meningitidis, another organism capable of causing urethritis. This may account for the discrepancy observed between Gram-positive findings and culture-positive results reported by the authors.

Response: Thank you for this comment. The utilised methods as described included use of sugar fermentation tests that are capable of differentiating between Neisseria species after culture enabling reporting only those confirmed to be N. gonorrhoeae.

Comment 3: Another important limitation—acknowledged by the authors—is the absence of data regarding patients’ sexual orientation. As a result, men who have sex with men (MSM) were not screened at all anatomical sites, which is a notable shortcoming. In this population, relying on urine samples alone is insufficient for comprehensive screening.

Response: Thank you for this comment. The study is based on a defined protocol of a surveillance program with specific inclusion criteria. We believe that non-inclusion of MSM limits generalizability, it is consistent with data suggesting that the major route of STI transmissions in sub-Saharan Africa is heterosexual. As any other study, there are always limitations and these ought to be declared as done in this case to inform result interpretation and application. This study also focuses on surveillance of drug resistance among people with gonorrhoea and we do not think there is documented differential risk of drug resistance by route of gonorrhea acquisition. Finally, our results provide baseline for future work design especially in case the MSM make part of the study or surveillance objectives. Also, no urine samples were considered during this scenario.

Comment 4: Additionally, I found the results of the multivariate analyses somewhat surprising. The lack of association between unprotected sex or multiple sexual partners and increased risk of gonorrhea is counterintuitive. The authors should provide further explanation or discuss potential confounding factors that could account for this unexpected finding.

Response: Thank you for this comment. Ideally, it would be expected that individuals who have unprotected sex or multiple sexual partners are at a higher risk for contracting gonorrhoea and testing positive. However, this is an association and not proved to be a direct causal relationship. Given that the reported results are of a sample of the population, it is possible to observe variations including a non-association due to lack of sufficient evidence statistically such as observed in this scenario. Further explanation has been added to the discussion as recommended currently reading as “..…. This could be a type II error due to the possibility of limited statistical power to detect this association.”. Please see line 176-180 in the updated manuscript.

Minor comments:

Comment 5: The statement that gonorrhea is “asymptomatic, especially in women” is poorly phrased. While this may be true for women, gonorrhea is predominantly symptomatic in men and the wording should reflect this contrast.

Response: Thank you for this comment. This has been improved in the revised manuscript.

Comment 6: Line spacing in the introduction is inconsistent and should be standardized for clarity and presentation.

Response: This has been improved. Thank you for the observation. 

Comment 7: Table 1 is difficult to interpret, particularly the percentages, which are unclear. I recommend reformatting the table to improve its readability and visual coherence.

Response: Thank you for this comment. The table is modified to define indirect values presented. Please see Table 1 in the updated manuscript.

Round 2

Reviewer 3 Report

Comments and Suggestions for Authors

The manuscript has been substantially improved after revision. Especially the figures are improved for readability. Also good to have data now in Table 1 on the N. gonorrhoeae prevalences in the different Kampala facilities and to have data that consider the results from culture.

There are however still some important obstacles, mainly in calculating and presenting the  percentages in two of the tables.

Please consider the points mentioned below and make proper adjustments.

Major comments.

  1. In Table 1 the patient characteristics are presented in percentages which are calculated ‘per class’. It should NOT be calculated relative to the number of persons with urethral symptoms since these numbers are not mentioned in Table 1. The percentages should be calculated relative to the number of persons with gonorroeae. For example in the column ‘Positive for N. gonorrhoeae (Gram staining) the HIV status should in all rows be calculated relative to the total number in that column (N= 1218). So HIV negative: N= 872 (and 872/1218 )= 72%; HIV positive: N= 78 ( and 78/1218) = 6.4%; HIV unknown: N=266 (and 266/1218 )= 21.8%.  The same goes for the column with the heading ‘Positive for N. gonorrhoeae (Gram staining and culture), in total N= 923. Please recalculate the percentages for ALL parameters (Age group, Number of sex partners, condom use, Facility name, etcetera)  in ALL rows to get more insightful data by calculating percentages relative to the number of N. gonorrhoeae positive as is given at the top of each column (N-1218 and N= 923).
  2. In Table 2 only percentages are presented. It is not clear how these were derived. What number is taken for 100%? Again here these numbers should be mentioned followed by percentages.
    For example: Gram positivity Overall (N=1663): N=1214 (73%) X- Y % (where X-Y is the 95% CI). Please add these numbers in table 2.
  3. Also the percentages in the Figures cannot be certified by the reader. What is the total number of cases (so: how many cases is 100%) for each division? Are these total numbers relating to the number of persons with symptomatic urethritis complaints? That should be clarified both in the text and in the Figure legend.

Author Response

Thank you very much for taking the time to review this manuscript. Please find the detailed responses below and the corresponding revisions in track changes in the re-submitted files. Point-by-point response to Comments and Suggestions for Authors The manuscript has been substantially improved after revision. Especially the figures are improved for readability. Also good to have data now in Table 1 on the N. gonorrhoeae prevalences in the different Kampala facilities and to have data that consider the results from culture. There are however still some important obstacles, mainly in calculating and presenting the percentages in two of the tables. Please consider the points mentioned below and make proper adjustments.

Major comments.

Comment 1: In Table 1 the patient characteristics are presented in percentages which are calculated ‘per class’. It should NOT be calculated relative to the number of persons with urethral symptoms since these numbers are not mentioned in Table 1. The percentages should be calculated relative to the number of persons with gonorroeae. For example in the column ‘Positive for N. gonorrhoeae (Gram staining) the HIV status should in all rows be calculated relative to the total number in that column (N= 1218). So HIV negative: N= 872 (and 872/1218 )= 72%; HIV positive: N= 78 ( and 78/1218) = 6.4%; HIV unknown: N=266 (and 266/1218 )= 21.8%. The same goes for the column with the heading ‘Positive for N. gonorrhoeae (Gram staining and culture), in total N= 923. Please recalculate the percentages for ALL parameters (Age group, Number of sex partners, condom use, Facility name, etcetera) in ALL rows to get more insightful data by calculating percentages relative to the number of N. gonorrhoeae positive as is given at the top of each column (N-1218 and N= 923).

Response: Thank you for this comment. The percentages have been recalculated with reference to the total positive cases per method (Column). Please see Table 1 in the revised manuscript.

Comment 2: In Table 2 only percentages are presented. It is not clear how these were derived. What number is taken for 100%? Again here these numbers should be mentioned followed by percentages. For example: Gram positivity Overall (N=1663): N=1214 (73%) X- Y % (where X-Y is the 95% CI). Please add these numbers in table 2.

Response: We appreciate this comment. The primary numbers for the cases have been added to the table in addition to the percentages and CIs for clarity. Please see Table 2 in the revised manuscript.

Comment 3: Also the percentages in the Figures cannot be certified by the reader. What is the total number of cases (so: how many cases is 100%) for each division? Are these total numbers relating to the number of persons with symptomatic urethritis complaints? That should be clarified both in the text and in the Figure legend.

Response: Thank you for this comment. The figures have been revised to include the respective total cases per division and the overall cases referred to in the captions below the figure. Please see Figure 1 and Figure 2 in the revised manuscript.

Reviewer 4 Report

Comments and Suggestions for Authors

The authors responded to the comments appropriately, resulting in an improvement in the quality of the manuscript.

Author Response

Thank you very much for taking the time to review this manuscript. Please find the detailed responses below and the corresponding revisions in track changes in the re-submitted files.

Point-by-point response to Comments and Suggestions for Authors

Comment 1: The authors responded to the comments appropriately, resulting in an improvement in the quality of the manuscript.

Response: Thank you for this comment. We again appreciate your time and technical contribution to the improvement of this manuscript.